# A preliminary study on assessment of wellbeing among veterinary medical house officers

**Munashe Chigerwe** [1]*, **Linda Barter**[2], **Julie E. Dechant**[2], **Jonathan D. Dear** [1], **Karen A. Boudreaux**[3]

**1** Department of Veterinary Medicine and Epidemiology, School of Veterinary Medicine, University of California Davis, Davis, California, United States of America, **2** Department of Surgical and Radiological Sciences, School of Veterinary Medicine, University of California Davis, Davis, California, United States of America, **3** Dean's Office, School of Veterinary Medicine, University of California Davis, Davis, California, United States of America

* mchigerwe@ucdavis.edu

**Data Availability Statement:** Data cannot be shared publicly because the data represents individual house officers' responses from a single institution (University of California Davis School of Veterinary Medicine). The data presented in our

## Abstract

Mental wellness is an important topic among practicing veterinarians. Peer reviewed studies focusing on veterinary house officers' wellbeing are lacking in veterinary medicine. The aim of this study was to assess wellbeing of house officers using validated surveys for anxiety, burnout, depression, and quality of life. A cross-sectional survey of 103 house officers (residents, interns, and fellows) was performed. Respondents were invited to voluntarily complete the online surveys. Anxiety, burnout, depression, and quality of life were assessed using the Generalized Anxiety Disorder (GAD-7), Maslach Burnout Inventory (MBI), Patient Health Questionnaire (PHQ-9), and Short Form-8 (SF-8), respectively. Descriptive statistics were calculated. For qualitative analysis, respondents were requested to rate their perception of the level of stress regarding various work-related stressors. The first survey was completed in 2017 with 60 respondents of which 51 (85%) identified as females and nine (15%) identified as males. The second survey was completed in 2018 with 43 respondents of which 35 (81.4%) identified as females and 8 (18.6%) identified as males. Respondents reported high levels of burnout characterized by high levels of emotional exhaustion and lack of personal accomplishment but reported mild levels of anxiety and depression. The mental component of their quality of life score was lower than the general US population, whereas the physical component score was consistent with the general US population. Respondents indicated moderate scores of stress for concerns regarding patient management, research, teaching, work-life balance, relationships, organizational skills, time management, finances, and the mental and emotional impact of the work environment. The high levels of burnout, and low mental quality of life in house officers require specific intervention programs to improve wellbeing.

## Introduction

Wellbeing among veterinarians is established as an area of concern for the veterinary profession in the United States [1–4] and other countries [5–9]. Factors that might increase the risk

study potentially contain sensitive identifying personal information such as mental health data which have psychological risks to the respondents. The minimal aggregated data set from the research are available from University of California Institutional Review Board Committee (contact via email provided below) for researchers who meet the criteria for access to confidential data. We have provided contacts (non-authors of this paper) to which data requests may be sent to obtain aggregated data. The contact is the institutional representative for the ethics committee: University of California Davis: Office of Research IRB Administration 1850 Research Park Drive Davis, CA 95618-6153 Phone: (530) 754-7679 Fax: (530) 754-7894 Email: ORExecutiveMgtAsst@ad3. ucdavis.edu.

**Funding:** The authors received no specific funding for this work.

**Competing interests:** The authors have declared that no competing interests exist.

for mental illness in veterinarians include personality traits such as the tendency to be over-achievers driven by perfectionism or neuroticism, and preference to work with animals rather than people that might lead to social isolation [5, 10, 11]. Work-related stressors such as long working hours, high client expectations, compassion fatigue, limited personal finances, possibility of client complaints and litigation, potential conflicts with peers [11], and being recently qualified [3, 12] have been suggested as other risk factors for mental illness in veterinarians. In addition to exposure to these risk factors, veterinary medical house officers (interns, residents, and fellows) might experience additional stressors including expectations to teach veterinary medical students, time constraints for studying for specialty board examinations, learning advanced clinical skills in their respective specialties, constant performance evaluation by supervisors and students, and designing and performing clinical research. Thus, it is anticipated that veterinary medical house officers might be at a high risk for anxiety, burnout, depression, and decreased quality of life.

## House officers program at the University of California Davis, Veterinary Medical Teaching Hospital

The House Officers Program officers at the University of California Davis Veterinary Medical Teaching Hospital offers a maximum of 48 training positions composed of 39 residency positions and 9 internships/fellowship, each year. The residency programs are 3–4 years in duration, whereas the internships/fellowships are 1-year long. The number of house officers hired each year varies with the specialty. Residency programs with a small animal focus include, behavior, cardiology, dentistry and oral surgery, dermatology, neurology/neurosurgery, oncology, ophthalmology, radiation oncology, emergency/critical care, internal medicine, surgery, and zoological companion animal medicine and surgery. Residency programs with a large animal emphasis include comparative theriogenology, dairy production medicine, equine surgery, livestock production and herd health, and internal medicine (livestock or equine emphasis). Support services residency programs include anatomic pathology, anesthesiology, clinical pathology, nutrition, and diagnostic imaging/radiology. Fellowships/internships programs include equine field services, equine integrative sports medicine, large animal ultrasound, marine mammal and pathology, and small animal emergency. The responsibilities of house officers include primary patient care including after hours emergencies, teaching veterinary students, and client communication. Additionally, residents are required to perform clinical research, and complete requirements for specialty boards examinations.

## Current mental health and wellness support for house officers

The University of California Davis campus through the Academic and Staff Assistance Program (https://www.hr.ucdavis.edu/departments/asap) offers cost-free services to house officers including mental health services that include managing anxiety and depression, self-care management including effective stress management, emotional intelligence, exercise, and work-related issues including assertiveness skills, bullying in the workplace, communication skills, and conflict resolution.

## Current study

Peer reviewed studies focusing on veterinary house officers' wellbeing are lacking in veterinary medicine. Studies on wellbeing of house officers are essential for program improvement by increasing awareness about mental health and providing wellness support. The objective of this preliminary study was to assess wellbeing of house officers using validated survey instruments (self-reported questionnaires) for anxiety, burnout, depression, and quality of life.

## Materials and methods

### Sampling and data collection

Sampling in 2017 and 2018 consisted of a non-probability, convenience, volunteer sample of house officers at the University of California Davis Veterinary Medical Teaching Hospital (VMTH). Email invitations to complete four survey instruments were sent with a link to an online survey tool (Qualtrics, Seattle, WA) through the secured VMTH portal allowing only invited house officers with appropriate login credentials to access the survey instruments. The email described the purpose of the study which was reiterated at the beginning of each survey instrument as that of gathering information to help improve the wellbeing of house officers. Each survey instrument also contained an electronic consent to participate in the study, as well as, the collection of demographic data, namely, gender, age, marital status, and dependents (spouse/partner, children, parents, grandparents, or other). Respondents were informed that the data collected for all instruments were confidential, and the anonymity of respondents was protected by removing all identifiable information before analysis. Questions were not presented in a forced choice format such that house officers could skip questions. The study was approved by the University of California-Davis, Institutional Review Board, which governs research with human participants.

### Survey instruments

The majority of internships, residency, and fellowship programs at the institution of study commence in August of each year. To increase the enrollment sample size, the survey was open for two cohorts of house officers, from August 2017–November 2017 and August 2018–November 2018. The validated survey instruments used to assess anxiety, burnout, depression, and quality of life were the following:

### Anxiety

The Generalized Anxiety Disorder (GAD-7) [13] is a self-administered questionnaire used as a screening tool for generalized anxiety disorders. The instrument is a 7-item survey, copyrighted (Pfizer Inc, New York, NY) but its usage is free [14]. The GAD-7 consists of 7 questions where house officers indicated the frequency with which they experienced designated problems. Scores were calculated by assigning a value of 0, 1, 2, or 3, to the response categories of 'not at all', 'several days', 'more than half the days', and 'nearly every day', respectively. The total score for each respondent was the sum of the scores for the seven items questions with a range from 0 to 21. Scores of 5–9 were considered indicative of mild anxiety, 10–14 moderate anxiety, and 15 or greater was considered severe anxiety. The internal consistency of the original GAD-7 instrument was excellent (Cronbach $\alpha$ = 0.92) [13].

### Burnout

Burnout is defined as a state of physical and mental exhaustion related to caregiving activities or work [15]. The Maslach Burnout Inventory (MBI) [16] was a 22-item survey, copyrighted (Mind Garden Inc, Menlo Park, CA) and used for this study through a contractual agreement to measure individual burnout levels through the subscales of emotional exhaustion, depersonalization, and personal accomplishment which are identified as the key aspects to burnout, thereby providing a context for why burnout could possibly have occurred [16]. The emotional exhaustion subscale assesses feelings of overextension and exhaustion from work with total scores of $\leq$ 18, 19–26 and $\geq$ 27 indicating low, average and high levels of burnout, respectively. The score for the emotional exhaustion subscale for each respondent was a sum of responses

to nine of the 22 items. The depersonalization subscale measures an impersonal response to other individuals (peers, students, staff, faculty or clients). The score for the depersonalization subscale for each respondent was a sum of responses to five of the 22 items. Depersonalization scores of $\leq 5$, 6–9 and $\geq 10$ indicated low, average and high levels of burnout, respectively. Total personal accomplishment subscale measures the house officer's feelings of competence and success. The score for the personal accomplishment subscale for each respondent was the sum of responses to eight of the 22 items. Personal accomplishment scores of $\geq 40$, 39–34 and $\leq 33$ indicated low, average, and high levels of burnout, respectively. The reliability of the original MBI as determined by Cronbach α were 0.89, 0.77, and 0.74 for the emotional exhaustion, depersonalization, and personal accomplishment subscales, respectively [16].

## Depression

The Patient Health Questionnaire (PHQ-9) [17] is a 9-item survey, copyrighted (Pfizer Inc, New York, NY) and free for use as a self-administered version of the PRIME-MD (Primary Care Evaluation of Mental Disorders) diagnostic instrument for common mental disorders, specifically focusing on depression to asked respondents to score nine depressive symptom criteria. The presence of each criterion during the previous 2 weeks was scored as 0 (not at all), 1 (several days), 2 (more than half the days), and 3 (nearly every day) resulting in a severity measure ranging from 0 to 27. Total score for each respondent was the sum of the scores from the nine items. Total scores of 0–4, 5–9, 10–14, 15–19, and 20–27, indicated mild, moderate, moderately severe, and severe levels of depression, respectively. The PHQ-9 is used to monitor the severity of depression and response to treatment as it is not a screening tool for depression. The internal reliability of the original PHQ-9 was excellent (Cronbach α = 0.89).

## Quality of life

Health related quality of life is a multi-dimensional construct that reflects a patient's perception of the impact of a disease and treatment on physical, psychological, and social function as related to overall well-being [18]. The Optum Short Form-8 Health Survey (SF-8) is an 8-item self-reported Likert scale instrument that assesses health related quality of life. The SF-8 has been established for the health domain scales of general health perception (GH), physical functioning (PF), role limitations due to physical health problems (role physical, RP), bodily pain (BP), energy/fatigue (vitality, VT), social functioning (SF), role limitation due to emotional problems (role emotional, RE), and psychological distress and well-being (mental health, MH) [19]. Two summary measures were determined, namely the physical component summary (PCS) and the mental component summary (MCS). Responses were standardized into scores for the eight dimensions using a commercial software made available through contractual agreement (Optum Pro CoRE, Eden Prairie, MN). Score of 50 points represent a reference average adult American population. Thus, scores below 50 correspond to deviations from normality and indicated a poorer quality of life, whereas scores above 50 points represented a better quality of life than that of the average adult American population [19]. Reliability of the original PCS and MCS were 0.88 and 0.82, respectively [19].

## Qualitative data collection on perception of level of stress

House officers were requested to rate their perception of the level of stress (score 1 = least stressful; score 5 = most stressful) regarding concerns about their patients, research, teaching, work-life balance, family-related issues, personal health, relationships, organizational skills, time management, financial situations, living environment, physical environment of the

VMTH, and mental and emotional impact of the environment of the VMTH. Additionally, house officers could make free text comments at end of the survey.

## Data analysis

Descriptive statistics for response rate, gender, marital status, age, and whether they had dependents were calculated. Normality check of the data was performed by the Shapiro-Wilk test. When data were normally distributed, mean ± standard deviation (SD) was reported, whereas median and interquartile range (IQR) was reported for not normally distributed data. Reliability (internal consistency) of each survey instrument was determined by calculating Cronbach's alpha (α). For the purposes of this study Cronbach's α for the subscales (emotional exhaustion, depersonalization and personal accomplishment) for burnout were used for analysis as they provide context for why burnout has occurred. The Cronbach α's for the mental component summary and physical component summary were used for analysis because they may guide specific intervention to improve the mental or physical components of quality of life. Scale Cronbach's α was used for analysis for the Generalized Anxiety Disorder and Patient Health Questionnaire instruments because both instruments do not have subscales. Cronbach's α of $\geq 0.7$ was considered to indicate acceptable reliability [20]. The scores for each instrument from 2017 were compared to that from 2018 using a Wilcoxon rank sum test. Multivariate analysis of variance (MANOVA) was performed for the combined data from 2017 and 2018 cohorts to determine the association between the scores for each instrument and selected explanatory variables. Based on previous studies that reported effect of different variables that affect levels of psychological stress in practicing veterinarians [3, 9], gender, marital status, and age were considered variables of interest. To ensure that MANOVA was appropriate for variables that were not normally distributed, distribution of the residuals and homogeneity of variances were assessed. To further confirm appropriateness of the MANOVA, analysis was performed using ranks for data points of variables that were not normally distributed. Correlation between the scores for each survey instrument were determined by calculation of Pearson's correlation coefficient ($r$). Commercial software (JMP Pro Version 15, SAS Institute Inc., Cary, NC) was used for analysis and $P < 0.05$ was considered significant. Perceptions on the level of stress for various work-related stressors were reported as median (range) scores. Content analysis was performed on free text comments.

## Results

### Respondents

The descriptive statistics including response rate, gender, age, marital status and dependents for the respondents is summarized in Table 1.

In our study, standardized reliability as indicated by Cronbach's α for MBI (burnout), GAD-7 (anxiety), PHQ-9 (depression) and SF-8 (quality of life) instruments were 0.78, 0.93, 0.90, and 0.87, respectively, and were considered acceptable. Cronbach's α for the MBI subscales, that is, emotional exhaustion, depersonalization, and personal accomplishment were 0.59, 0.60, and 0.88, respectively. Cronbach's α for SF-8 subscales of mental component summary and physical component summary were 0.87 and 0.85, respectively, in our study.

### Anxiety (GAD-7), burnout (MBI), and depression (PHQ-9)

Median (IQR) GAD-7 scores for anxiety were 5 (8.3), 5.5 (9), and 5 (9) in 2017, 2018, and combined data for the two years, respectively, indicating mild levels of anxiety. There was no difference ($P = 0.922$) in the GAD-7 anxiety scores in 2017 and 2018. Median (IQR) scores for

**Table 1. Descriptive statistics of University of California Davis house officers in 2017 and 2018.**

|  | 2017 | 2018 |
|---|---|---|
| Response rate | 34.3% (60/175) | 36% (43/119) |
| Gender |  |  |
| Female | 85% (51/60) | 81.4% (35/43) |
| Male | 15% (9/60) | 18.6% (8/43) |
| *Age |  |  |
| 26–30 | 59.3% (35/59) | 55.8% (24/43) |
| 31–35 | 35.6% (21/59) | 34.9% (15/43) |
| 36–40 | 5.1% (3/59) | 9.3% (4/43) |
| Marital status |  |  |
| Single | 73.3% (44/60) | 72.1% (31/43) |
| Married | 26.7% (16/60) | 23.3% (10/43) |
| Engaged | 0 | 4.6% (2/43) |
| Dependents | 6 | 2 |

*One respondent did not indicate their age in 2017.

comparisons between 2017 and 2018 for MBI are summarized in Table 2. The MBI scores revealed high levels of burnout in both 2017 and 2018. The scores for emotional exhaustion ($P = 0.636$), depersonalization ($P = 0.853$), and personal accomplishment ($P = 0.434$) were not different between 2017 and 2018. Median (IQR) for PHQ-9 scores were 7 (8.5), 7.5 (10.5) and 7 (9.8) for 2017, 2018, and combined data for the two years, respectively, indicating mild levels of depression.

Gender (model $P = 0.0412$) was a significant predictor of emotional exhaustion scores for the MBI, with higher scores reported in males compared to females (33.8 *vs* 26.9; $P = 0.035$). Gender was a significant predictor of PHQ-9 scores, with higher scores reported in males compared females (11.5 *vs* 7.8; $P = 0.027$). Gender was also a predictor of GAD-7 scores with higher scores reported in males compared to females (10.1 *vs* 6.9; $P = 0.040$).

Marital status (model $P < 0.0001$) was significant predictor of emotional exhaustion (32.4 *vs* 26.5; $P < 0.033$) and depersonalization (11.2 *vs* 7.5; $P = 0.011$) scores of the MBI (burnout) with higher scores reported in married compared to single respondents for both subscales. Marital status was also a predictor of personal accomplishment subscale of MBI with higher scores reported in single compared to married respondents (32.8 *vs* 28.6; $P = 0.010$). Marital status was also a predictor of mental component summary subscale of the quality of life (Short

**Table 2. Comparison of scores (median and interquartile range) of burnout with Maslach Burnout Inventory (MBI) for University of California Davis house officers in 2017 and 2018.**

|  | Year 2017 | Year 2018 | Combined 2017 and 2018 | *P*-value |
|---|---|---|---|---|
| Emotional exhaustion (EE) | 29.5 (3–50) | 27 (7–51) | 27 (19.5) | 0.636 |
| Depersonalization (DP) | 7 (0–26) | 6.5 (0–23) | 7 (9) | 0.853 |
| Personal accomplishment (PA) | 32 (13–44) | 33 (19–47) | 32 (10) | 0.434 |
| Number of respondents | 60 | 43 | 103 |  |

EE scores of $\leq 18$, 19–26 and $\geq 27$ indicate low, average, and high levels of burnout, respectively.

DP scores of $\leq 5$, 6–9 and $\geq 10$ indicate low, average, and high levels of burnout, respectively.

PA scores of $\geq 40$, 39–34 and $\leq 33$ indicate low, average, and high levels of burnout, respectively.

Row score comparisons with $P < 0.05$ between 2017 and 2018 are different.

Form-8) with higher scores reported in single compared to married respondents (42.5 *vs* 31.2; *P*<0.0001). Furthermore, marital status was a predictor of depression (PHQ-9) and anxiety scores (GAD-7) with higher scores reported for married compared to single respondents for PHQ-9 (12.0 *vs* 7.2; *P* = 0.0006) and GAD-7 (12.5 *vs* 5.8; *P*<0.0001).

Emotional exhaustion subscale of the MBI scores were positively correlated with PHQ-9 ($r$ = 0.59) and GAD-7 scores ($r$ = 0.50) but negatively correlated ($r$ = -0.52) with mental component summary scale (MCS) scores of the SF-8 (quality of life). Depersonalization subscale of the MBI scores were positively correlated with PHQ-9 ($r$ = 0.47) and GAD-7 scores ($r$ = 0.54) but negatively correlated with MCS ($r$ = -0.51). Personal accomplishment subscale of the MBI scores was negatively correlated with PHQ-9 ($r$ = -0.49) and GAD-7 scores ($r$ = -0.44) but positively correlated with MCS ($r$ = 0.40). The MCS scores were negatively correlated with PHQ-9 ($r$ = -0.72) and GAD-7 scores ($r$ = -0.67). The GAD-7 and the PHQ-9 scores were positively correlated ($r$ = 0.76). Age was not a significant predictor of scores for any of the survey questionnaires (*P* = 0.405).

## Quality of life

Quality of life scores for 2017 and 2018 are summarized in Table 3. All mean scores for the 8 dimensions were below 50, except RP and BP dimensions in 2017. Scores for RE, MH, VT, GH, RP, PF, and BP were not different (*P* >0.05) between 2017 and 2018 (Table 3). The SF dimension was lower (*P* = 0.046) in 2018 (41.4 ± 8.3) compared to 2017 (45.0 ± 9.6), but both scores were below 50. The MCS was below 50 for 2017 and 2018, whereas the PCS was above 50 for 2017 and 2018. The MCS and PCS score were not different (*P* >0.05) between 2017 and 2018 (Table 3).

Scores below 50 points corresponded to deviations from normality and indicate a poorer quality of life, whereas scores above 50 points represented a better quality of life than that of the average adult American population [19]. Row score comparisons with *P* < 0.05 between 2017 and 2018 are different.

## Quantitative and qualitative data collection on perception of level of stress

Summary of perception of level of stress among house officers are summarized in Table 4. For both years, respondents indicated similar moderate scores of stress for concerns regarding patient management, research, teaching, work-life balance, relationships, organizational skills, time management, finances, and the mental and emotional impact of the work environment of the VMTH.

**Table 3. Comparison of quality of life dimensions scores (mean ± standard deviation) with Short Form-8 (SF-8) scores among University of California Davis house officers in 2017 (n = 60) and 2018 (n = 43).**

|  | Year 2017 | Year 2018 | Combined 2017 and 2018 | *P*-value |
|---|---|---|---|---|
| Role emotional (RE) | 44.6 ± 7.8 | 42.4 ± 7.9 | 43.7 ± 7.9 | 0.162 |
| Mental health (MH) | 42.4 ± 10.3 | 40.2 ± 10.2 | 41.5 ± 10.3 | 0.297 |
| Social functioning (SF) | 45.0 ± 9.6 | 41.4 ± 8.3 | 43.5 ± 9.2 | 0.046 |
| Vitality (VT) | 46.0 ± 8.7 | 45.4 ± 6.9 | 45.8 ± 8.0 | 0.736 |
| General health perception (GH) | 45.2 ± 7.0 | 43.6 ± 8.0 | 44.5 ± 7.4 | 0.276 |
| Role physical (RP) | 50.1 ± 5.3 | 48.9 ± 5.3 | 49.6 ± 5.3 | 0.262 |
| Physical functioning (PF) | 48.7 ± 7.6 | 49.1 ± 5.8 | 48.9 ± 6.8 | 0.751 |
| Bodily pain (BP) | 51.3 ± 7.4 | 49.9 ± 6.6 | 50.7 ± 7.1 | 0.341 |
| Mental component summary (MCS) | 41.1 ± 12.5 | 37.9 ± 12.1 | 39.8 ± 12.4 | 0.196 |
| Physical component summary (PCS) | 51.2 ± 6.3 | 50.4 ± 6.9 | 50.9 ± 6.5 | 0.550 |

**Table 4. Median (range) scores for perception of stress regarding various aspects of training among house officers in 2017 (n = 60) and 2018 (n = 43).**

| Question | 2017 | 2018 |
|---|---|---|
| Concerns about patient management | 4 (1–5) | 4 (1–5) |
| Concerns about my research | 3 (1–5) | 3 (1–5) |
| Concerns about my teaching | 3 (1–5) | 3 (1–5) |
| Work/life balance | 3 (1–5) | 3 (1–5) |
| Family-related issues | 2 (1–5) | 2 (1–5) |
| Personal health | 2 (1–5) | 3 (1–5) |
| Relationships | 3 (1–5) | 3 (1–5) |
| Organizational skills | 3 (1–5) | 3 (1–5) |
| Time management | 3 (1–5) | 4 (1–5) |
| Financial situation | 3 (1–5) | 3 (1–5) |
| Living environment | 2 (1–4) | 2 (1–5) |
| Physical environment of the VMTH | 2 (1–5) | 2 (1–5) |
| Mental and emotional impact of the environment of the VMTH | 3 (1–5) | 3 |

The stem of the question was framed as:'From least stressful to most stressful, please rate the following with regards to your life as an employee of the VMTH:'

Scores range from 1 (least stressful) to 5 (most stressful).

VMTH—Veterinary Medical Teaching Hospital.

Content analysis from free text comments from respondents indicated that financial instability, lack of collegiality among supervising faculty, lack of appropriate feedback from supervising faculty, worry about employment opportunities after graduation from programs, difficult client interactions, lack of mentoring by faculty supervisors, low quality of mental health, and coping with high academic demands of training were significant sources of stress.

## Discussion

The findings in this study suggest that veterinary house officers consistently experienced high levels of burnout as indicated by personal accomplishment and emotional exhaustion scores. Findings further suggest that burnout might be related to stressors such as lack of time management and organizational skills, lack of research and teaching skills, financial instability, lack of balancing work and non-work activities, and undesirable working environment in sections of the VMTH. While there are no comparable studies in veterinary medicine, causes of burnout in medical residents include sleep deprivation, high workload, unsatisfactory salaries, [21] and taking many responsibilities at work [22, 23]. The consequences of burnout in medical residents that are applicable to veterinary house officers include increased risk for medical errors, adverse effects on patient safety, depression [15], and suicidal ideation [24]. Burnout results from an individual being overstretched in their work place. Therefore, if the source of burnout is removed or reduced, the stress experienced by an individual is reduced whereas depression affects one's holistic life (work and outside work). Burnout can predispose to depression and in our study, the levels of burnout require attention.

High levels of burnout in medical residents were associated with high levels of anxiety during medical school [25]. In contrast, participants in our study reported mild levels of anxiety, which is desirable, despite high levels of anxiety reported in North American veterinary students [26, 27]. While participants in our study reported mild levels of depression, surveys in the United States indicated that veterinarians experience high levels of depression and suicidal

ideation compared to the US general population [1–4]. These differences in anxiety and depression levels suggest that the structure of a large house officer training program might create a support network of like-minded peers that may alleviate some of the stressors experienced by veterinary students and practicing veterinarians, consistent with previous studies [3]. Furthermore, house officers may be protected from the daily stressors of veterinary practice because they are in a supervised training programs that allow for protected time for non-clinical activities. However, these aspects of house officer training do not necessarily result in durable coping strategies or resilience and might not be transferrable to post-training working environment. Therefore, it is important to note that house officers will join the veterinary workforce after graduating and will be exposed to the stressors associated with high levels of depression among veterinarians, which may result in deterioration of their mental health and wellbeing.

Assessment of quality of life allows identification of the most compromised dimensions of well-being and therefore, the ability to intervene by providing resources for house officers. Quality of life assessments allow comparison of the house officers in our study to a reference United States adult population, and therefore are interpreted as deviations from normality. Participants in our study reported lower quality of mental health as indicated by low scores for role emotional (RE), mental health (MH), social functioning (SF), vitality (VT), general health perception (GH) and the mental component summary (MCS). This result might be related to the work-related stressors indicated by the results of the qualitative analysis. In contrast, house officers in our study had physical component summary scores >50 consistent with their relatively young age (26–35 years). This is because young adults are likely to experience better physical health compared to older individuals, and previous studies have reported that physical component summary scores decrease with advanced normal aging [28].

In our study male respondents had higher scores for emotional exhaustion, PHQ-9 (depression) and GAD-7 scores (anxiety) compared to female respondents. This result contrasts with previous studies that reported higher risk for psychological distress for female veterinarians compared to males [3, 7, 9]. However, it should be noted that female respondents were over-represented (83.5%) in our study, and therefore our study results should be interpreted with caution. Married respondents reported higher scores for all subscales of burnout, GAD-7, and PHQ-9, but lower scores for the mental component summary subscale for SF-8 (quality of life) suggesting higher psychological distress compared to unmarried respondents. A possible reason for this finding might be increased additional stress associated with managing a family among married respondents. Previous studies indicated that negative at-home interference such as communication with clients after office hours potentially predisposes to strained spousal relationships for married respondents compared to unmarried respondents [29, 30]. In contrast, other studies have reported a protective psychological of effect of marriage [31]. It should be noted that in our study unmarried respondents were overrepresented (75%) and therefore our results should be interpreted with caution. Age was not a significant predictor of survey questionnaire scores in our study, consistent with previous studies that indicated that poor mental health outcomes in young veterinarians is a combination of other personal and career attributes [5]. Correlations and direction of the correlations (negative or positive) among scores of the various instruments when gender or marital status were expected.

The results of our study are a basis for improvements and intervention of training programs to reduce burnout and improve wellbeing among house officers. While burnout might be inevitable among house officer training programs, awareness about burnout is crucial to short-term and long-term improvements of these training programs. Awareness regarding burnout can be disseminated during program orientation or during performance evaluation of house officers. A significant proportion of veterinarians have been reported to have negative attitudes

towards treatment and social support for mental illness [32] and might not seek help when required. Therefore, awareness of burnout for house officers should include availability of resources, and recommendations for house officers to consider accessing the resources. Specific examples of resources include a veterinary practice house officer ombudsman/woman, and on-campus counselling services. The VMTH at our institution appointed a faculty member (Director of House Officer Affairs and Education) to collaborate with a Graduation Clinical Education Committee to resolve concerns regarding house officers. Furthermore, house officers might require representation of their training related concerns at the VMTH level through 2–4 elected (depending on the size of the program) chief house officers.

The personal accomplishment subscale of burnout assessed feelings of competence and successful achievement in a house officer's work. Personal accomplishment and depersonalization determine emotional exhaustion [33]. Consequently, focusing on early signs of emotional exhaustion is not recommended because when the signs of emotional exhaustion appear, the burnout process is already underway [33]. The high scores of emotional exhaustion, low scores of personal accomplishment, lack of collegiality among faculty, and lack of appropriate feedback from faculty reported by house officers indicate the need for improvement in interactions between house officers and faculty to create a safe learning environment. To increase the sense of personal accomplishment and counteract burnout, specific training programs have been recommended for employees in various fields [34]. These training programs include role-playing to provide success experiences (enactive mastery), models of performances (vicarious experiences), coaching and encouragement (verbal persuasion) [34]. Specifically, veterinarians supervising house officers should behave as role models when communicating with other veterinarians or clients, provide constructive feedback, and effectively mentor house officers on research and teaching skills. House officers have variable teaching experience, and respondents in our study identified teaching students as an important stressor. Therefore, basic interactive courses with application exercises (either in online or in-person format) on how students learn should be incorporated in the training to improve house officers' teaching skills and confidence.

Further long-term approaches to reduce burnout for house officers include incorporation of mindfulness-based stress reduction (MBSR) intervention programs during training [35]. Mindfulness-based stress reduction intervention programs are structured programs that employ mindful meditation to alleviate suffering associated with physical, psychosomatic, and psychiatric disorders [35]. The programs are non-religious and non-esoteric and help individuals develop enhanced awareness of mental experiences [35]. Consequently, this improves an individual's self-care and well-being, including during periods of high stress [36].

Limitations of our study include the small sample size, from a single institution and therefore the results have limited external validity and cannot be generalized to house officers other than those enrolled in the study. Participation in the study was voluntary and may introduce response bias as house officers with an interest in the topic might have self-selected to complete the surveys. The surveys consisted of self-reported data to which social desirability bias may be present as participants may respond to items in what they feel are socially acceptable selections. Results from self-reported data are also dependent on the participants introspective ability, honesty, and ability to interpret the question which leads to further limitations of our study. With a volunteer sample, sampling bias may also be present as personal motivations for completing the instruments was unknown. Due to the small sample size, our study did not assess influence of the training stage (for instance, Year 1 compared to Year 2 of training) on wellbeing of house officers. Overrepresentation of females in our house officers study population is consistent with a relative higher proportion of practicing female veterinarians [3]. Future studies should consider cross-sectional studies evaluating wellbeing of house officers in

different programs across North America. In medical residencies, burnout was significantly higher in surgical/urgency than in other clinical specialties [37]. Urology, neurology, emergency medicine, and general surgery were also associated with higher risks of burnout relative to training in internal medicine [25]. Therefore, further studies should also evaluate influence of training stage and specialty on wellbeing of house officers. Our study only focused on factors associated with psychological stress but not aspects of house officers' training that improve mental health and wellness. Future studies should focus on surveys such as measure of subjective happiness [38] that evaluate factors that positively enhance mental health and wellness among house officers during training.

## Conclusion

Veterinary house officers experience high levels of burnout characterized by high levels of emotional exhaustion and lack of personal accomplishment, and mild levels of anxiety and depression. The mental component of quality of life of house officers is lower than the general US population, whereas the physical component of their quality of life is consistent with the general US population. Patient management, research, teaching, work-life balance, relationships, organizational skills, time management, finances, and the mental and emotional impact of the work environment were qualitatively identified as important stressors affecting wellbeing.

## Acknowledgments

The authors thank Dr. Kate Hopper, Jan Harlan, and Nicole Adams for their assistance in data collection. The authors thank all house officers who completed the surveys.

## Author Contributions

**Conceptualization:** Munashe Chigerwe, Karen A. Boudreaux.

**Data curation:** Karen A. Boudreaux.

**Formal analysis:** Munashe Chigerwe, Karen A. Boudreaux.

**Investigation:** Linda Barter.

**Methodology:** Munashe Chigerwe.

**Project administration:** Munashe Chigerwe.

**Resources:** Linda Barter, Julie E. Dechant, Jonathan D. Dear, Karen A. Boudreaux.

**Supervision:** Munashe Chigerwe, Karen A. Boudreaux.

**Writing – original draft:** Munashe Chigerwe.

**Writing – review & editing:** Munashe Chigerwe, Linda Barter, Julie E. Dechant, Jonathan D. Dear, Karen A. Boudreaux.

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
