## [Decision Letter · Decision Letter 0]

28 Oct 2020

PONE-D-20-17459

A preliminary study on assessment of wellbeing among veterinary medical house officers

PLOS ONE

Dear Dr. Chigerwe,

Thank you for submitting your manuscript to PLOS ONE. After careful consideration, we feel that it has merit but does not fully meet PLOS ONE’s publication criteria as it currently stands. Therefore, we invite you to submit a revised version of the manuscript that addresses the points raised during the review process.

We look forward to receiving your revised manuscript.

Kind regards,

Serge Brand

Academic Editor

PLOS ONE

Journal Requirements:

3. In your Methods section, please provide additional information about the participant recruitment method and the demographic details of your participants. Please ensure you have provided sufficient details to replicate the analyses such as: a) the recruitment date range (month and year), b) a description of any inclusion/exclusion criteria that were applied to participant recruitment, c) a table of relevant demographic details, d) a statement as to whether your sample can be considered representative of a larger population, e) a description of how participants were recruited, and f) descriptions of where participants were recruited and where the research took place.

4.We note that you have indicated that data from this study are available upon request. PLOS only allows data to be available upon request if there are legal or ethical restrictions on sharing data publicly. For information on unacceptable data access restrictions, please see http://journals.plos.org/plosone/s/data-availability#loc-unacceptable-data-access-restrictions.

Additional Editor Comments (if provided):

Dear Authors,

As you'll see, three Reviewers and experts in the field commented on your manuscript. While they identified some merits, they also raised major concerns. Herewith, I do invite you to try to cope with the Reviewers' concerns. When re-submitting a revision, please ensure to use the track change mode. Note that asking for a revision does not imply an acceptance.

Sincerely,

Serge Brand

Reviewers' comments:

Reviewer's Responses to Questions

**Comments to the Author**

1. Is the manuscript technically sound, and do the data support the conclusions?

Reviewer #1: Yes

Reviewer #2: No

Reviewer #3: Partly

2. Has the statistical analysis been performed appropriately and rigorously? 

Reviewer #1: Yes

Reviewer #2: No

Reviewer #3: No

3. Have the authors made all data underlying the findings in their manuscript fully available?

Reviewer #1: Yes

Reviewer #2: No

Reviewer #3: Yes

4. Is the manuscript presented in an intelligible fashion and written in standard English?

Reviewer #1: Yes

Reviewer #2: Yes

Reviewer #3: Yes

5. Review Comments to the Author

Reviewer #1: Thank you for the opportunity to review the article. This topic has become more warranted given more attention is paying for the wellbeing of people who in veterinary and animal welfare profession. I have got a few queries and comments for the authors to consider and address:

1. The authors sampled two cohorts of participants in 2017 and 2018. I would like the authors to provide a bit more information (aside from what was written on p. 4, line 92 to 94) on why they recruited in these two different period. Was it because they mainly just want to increase the sample size for analysis?

2. Aside from the authors reporting the Cronbach Alphas for the measures from their own study in the results section, I suggest they also report the measures' initial Cronbach Alphas from the original studies in the Instrument sections.

3. I am a bit confusing with the "Qualitative data collection on perception of level of stress". It looks to me (based on the results) that a scale was used to detect the perception but then there was a free text comments for participants to write qualitative comments in relation to stress. When I read the results, it looks like the authors just reported numbers that matched categories, rather than really engaging with the qualitative data to illustrate the content. This raised the question to me whether the identification of themes provided was mainly based on content analysis. I think the authors need to provide a bit more discussion and distinction. My original thought was that aside from identifying themes from the free texts, there will be some captions be used to further illustrate those themes but that does not seem to be the case. I would probably reframe it as using the score to identify perception of level of stress. Free text enables participants to provide more in-depth descriptions and those descriptions were sorted by content analysis. The way it is being described here would have made other readers to assume some qualitative comments be illustrated to further support the descriptive data.

Reviewer #2: 1. Thank you for the opportunity to review this submission.

2. The paper reports on a small-scale cross-sectional survey of well-being among veterinary medical house officers at a training organization in the USA. It would be helpful to have more context, e.g. the responsibilities, support and duration of a veterinary house officer’s role, as not all training programs use this structure.

3. Data and data analysis

4. Data were collected by a cross-sectional study with 103 respondents, who completed online surveys on anxiety, burnout, depression and quality of life, using validated published scales. The data were collected in 2017 (N=60, 85% female, 34% response rate) and 2018 (43 respondents, 81% female, 36% response rate).

5. Some points to clarify:

6. On p. 16 we learn that it’s a 2-year training program so it is important to clarify whether participants in each year were different cohorts or the same people surveyed twice (a repeated measures design).

7. Participants were asked to rate, not rank, their perceptions of stress associated with the listed stressors, and this is quantitative, not qualitative, data.

8. The open-ended questions do provide qualitative data; there is no information on how this was analyzed (e.g. thematic analysis).

9. ‘Selected independent variables’ are mentioned on p. 7 but it’s not clear how they were selected or what they or the dependent/criterion variables were. There is no hypothesis-testing.

10. P. 7 states that demographic variables were included; p. 9 states that they were omitted. Without these analyses, there is no information on which groups, if any, were more affected.

11. Cronbach’s alpha is given for the scales overall but is also required for each sub-scale, as the analyses were based on these.

12. There is insufficient information on how the quality of life scores were derived. The burnout measures need number of items and response options.

13. The statistical analyses compare scores between the 2017 and 2018 groups. However, there is no testing of sample vs. population distributions.

14. Findings are over-stated. For example, conclusions are made about overall high levels of burnout when scores are on, or within a point of, a threshold and there is wide variation in the data.

15. The list of stressors with ‘relatively higher scores’ includes nearly all of them (10 of 13), and 8/13 score 3 on the 5 point scale. As scores are rounded to whole numbers this suggests most were at the scale midpoint, not ‘high’ scores.

16. Overall, the data do not give strong evidence of problems throughout the cohort. More interestingly perhaps, variation in the data suggests that some individuals may be experiencing difficulties, so targeted solutions matched to individual needs may be required.

17. Conclusions

18. The paper focuses on problems and argues for interventions but the evidence is thin. There is little said about the null findings for several variables or that, where others exceeded cut-offs, it was never by much. There are considerable problems – logical and statistical – with saying that a score of 49.1 is fine but 50.1 is problematic, for instance.

19. As the PHQ is described, it’s not possible to be ‘not at all’ depressed; at best people have to be ‘mildly’ depressed. Depression is a serious diagnosable mental health condition, so telling participants that they are ‘mildly depressed’ based on these scores could cause problems in itself.

20. Claims that participants ‘consistently experienced high levels of burnout’ do not match the data, from which burnout does not appear to be extreme or widespread.

21. The authors suggest that ‘burnout was associated with stressors such as…’ but no tests of association (e.g. correlation) are presented.

22. Conclusions about possible intervening variables - coping and resilience, and wider implications e.g. transfer to working environments, are not part of the study.

23. Although there is little evidence of problems, the wider literature is used to argue that there are in fact problems which need to be solved. Scale ranges suggest that some participants may be experiencing problems but this does not show which interventions are required or by whom. Targeted approaches focused on needs would seem preferable. The suggested solutions are not based on the study findings.

24. Clarity

25. It’s well written and readable, except that the use of acronyms in the results section means constantly referring back to what each one means. There’s also a constant need to refer back to information on score thresholds to work out what mean scores indicate.

26. Summary

27. Overall, then, this is a small-scale study in one training institution which does not allow conclusions to be drawn about a profession, or even veterinary training, as a whole. The findings are over-stated and focus too much on the negative, and the suggested interventions are unrelated to the study findings. While professional well-being in veterinarians is interesting and important, the conclusions should be carefully aligned with the findings.

Reviewer #3: Comments for the Authors:

Thank you for the opportunity to review your manuscript. Wellbeing of veterinarians is an important scientific topic and deserves to be investigated.

After reading your manuscript one is confused what the purpose was of carrying out the study in two consecutive school-years? What made you think that veterinarians’ wellbeing may be changed in one year? What happened in that period that might approve this study design? Another question is how many people from the 2017 and 2018 samples are actually the same persons? We don’t know anything about that after reading your description of the results? Further, as the samples from 2017 and 2018 are not dependent nor independent but mixed, there is no statistical test that might compare the respondents wellbeing in 2017 and 2018. In conclusion, you have chosen a wrong method to fulfill the purpose of the study. Therefore, unfortunately I have to recommend the editor to reject your paper.

6. PLOS authors have the option to publish the peer review history of their article (what does this mean?). If published, this will include your full peer review and any attached files.

Reviewer #1: No

Reviewer #2: No

Reviewer #3: No

---

## [Author Response · Author response to Decision Letter 0]

15 Mar 2021

Reviewer #1: 

Thank you for the opportunity to review the article. This topic has become more warranted given more attention is paying for the wellbeing of people who in veterinary and animal welfare profession. I have got a few queries and comments for the authors to consider and address:

AU: Thank you for taking time to review our manuscript. Below is point-by-point response to your comments. To make it easier for you, we highlighted responses to your comments in green.

1. The sampled two cohorts of participants in 2017 and 2018. I would like the authors to provide a bit more information (aside from what was written on p. 4, line 92 to 94) on why they recruited in these two different period. Was it because they mainly just want to increase the sample size for analysis?

AU: Yes, that is correct that we wanted to increase the sample size. This was because data analysis of one of the instruments (Quality of life scores which requires standardization of responses using a special software) for the 2017 cohort was completed in 2018. Therefore, to increase the sample size we enrolled the second group of house officers in 2018.

In response to your comment, we added a comment indicating that another reason for enrolling 2 cohorts was to increase sample size. Please see Materials and Methods under “Survey instruments”.

2. Aside from the authors reporting the Cronbach Alphas for the measures from their own study in the results section, I suggest they also report the measures' initial Cronbach Alphas from the original studies in the Instrument sections.

AU: We included the Cronbach’s alphas for each instrument. Please see Materials and Methods under “Survey instruments”.

3. I am a bit confusing with the "Qualitative data collection on perception of level of stress". It looks to me (based on the results) that a scale was used to detect the perception but then there was a free text comments for participants to write qualitative comments in relation to stress. When I read the results, it looks like the authors just reported numbers that matched categories, rather than really engaging with the qualitative data to illustrate the content. This raised the question to me whether the identification of themes provided was mainly based on content analysis. I think the authors need to provide a bit more discussion and distinction. My original thought was that aside from identifying themes from the free texts, there will be some captions be used to further illustrate those themes but that does not seem to be the case. I would probably reframe it as using the score to identify perception of level of stress. Free text enables participants to provide more in-depth descriptions and those descriptions were sorted by content analysis. The way it is being described here would have made other readers to assume some qualitative comments be illustrated to further support the descriptive data.

AU: Thank you for your comment. Thematic analysis was not performed in this case. And as you indicate, analysis of the free text is simply content analysis. We phrased to indicate that content analysis instead of thematic analysis was performed. Please see Data Analysis and Results (qualitative data collection of perception of levels of stress).

Reviewer #2: 1. Thank you for the opportunity to review this submission.

AU: Thank you for taking time to review our manuscript. We responded point-by-point to your comments. To make it easier for you to find responses specific to your comments, we highlighted the responses in yellow.

2. The paper reports on a small-scale cross-sectional survey of well-being among veterinary medical house officers at a training organization in the USA. It would be helpful to have more context, e.g. the responsibilities, support and duration of a veterinary house officer’s role, as not all training programs use this structure.

AU: Thank you. We added more context regarding the house officers program including responsibilities, duration, and support opportunities. Please see the Introduction.

3. Data and data analysis

4. Data were collected by a cross-sectional study with 103 respondents, who completed online surveys on anxiety, burnout, depression and quality of life, using validated published scales. The data were collected in 2017 (N=60, 85% female, 34% response rate) and 2018 (43 respondents, 81% female, 36% response rate).

5. Some points to clarify:

6. On p. 16 we learn that it’s a 2-year training program so it is important to clarify whether participants in each year were different cohorts or the same people surveyed twice (a repeated measures design).

AU: We clarified this in the following ways:

1. We included the description of the House Officers Program, including the length of the residency programs and fellowships/internships (Please see the Introduction). Whereas the fellows/interns respondents were different in 2017 and 2018 because they are 1 year-programs, some of residents (unless they were in their final year of the residency program) potentially responded to the surveys in both 2017 and 2018. We did not track the residents who might have responded to the survey in 2017 and 2018 to maintain anonymity of the respondents. 

2. We also included both 2017 and 2018 in our study to increase our sample size. We indicated this in the Materials and Methods, under the subsection “Survey instruments”. The changes are highlighted in green as Reviewer #1 raised similar concerns.

7. Participants were asked to rate, not rank, their perceptions of stress associated with the listed stressors, and this is quantitative, not qualitative, data.

AU: Replaced the word “rank” with “rate” the Abstract and Materials and Methods. 

8. The open-ended questions do provide qualitative data; there is no information on how this was analyzed (e.g. thematic analysis).

AU: Thank you. Content analysis rather than thematic analysis was performed. We corrected this in the Materials and Methods (Data analysis) and Results section. The changes are highlighted in green as Reviewer #1 raised similar concerns.

9. ‘Selected independent variables’ are mentioned on p. 7 but it’s not clear how they were selected or what they or the dependent/criterion variables were. There is no hypothesis-testing.

AU: We included the reasons why the chosen independent variables were considered and included relevant references. Please see Data Analysis section. 

As indicated in our first submission, age was distributed across the various strata considered in our study, but respondents who identified as female and unmarried respondents were overrepresented in our population. We included the data of the multivariate analysis. We discussed the significant variables predicting survey scores, namely gender and marital status. We also discussed age as a variable as it was not a significant predictor of survey scores. We also indicated in the discussion that this should be interpreted with caution due to female and unmarried respondents being overrepresented. Please see results section and discussion. We also included appropriate references in the discussion.

10. P. 7 states that demographic variables were included; p. 9 states that they were omitted. Without these analyses, there is no information on which groups, if any, were more affected.

AU: Please see clarification and comments above. 

11. Cronbach’s alpha is given for the scales overall but is also required for each sub-scale, as the analyses were based on these.

AU: We provided the scales Cronbach’s alpha for all survey instruments in the Materials and Methods, including the subscales for the Maslach Burnout Inventory and Short-Form 8 (Quality of life). We also provided Cronbach’s alpha for the Maslach Burnout Inventory, which has 3 subscales, and Short-Form -8 scale which has 2 subscales, in the results. Please see results. The other survey instruments, namely Generalized Anxiety Disorder, and Patient Health Questionnaire -9 (Depression) only have items without subscales. 

12. There is insufficient information on how the quality of life scores were derived. The burnout measures need number of items and response options.

AU: Both the Quality of Life (SF-8) and Maslach Burnout Inventory survey instruments are copyrighted. Legally we cannot provide the items. We had access to the instruments through a paid contract. The GAD-7 and PHQ-9 are currently available for use in research and can be accessed online. As indicated in our first submission, we provided as much information as we can without violating any copyrights. We included the number of items for SF-8 and Maslach Burnout Inventory.

13. The statistical analyses compare scores between the 2017 and 2018 groups. However, there is no testing of sample vs. population distributions.

AU: We included information of testing distribution in the Data analysis section.

14. Findings are over-stated. For example, conclusions are made about overall high levels of burnout when scores are on, or within a point of, a threshold and there is wide variation in the data.

AU: Our interpretation of high burnout was based on higher mean scores of emotional exhaustion and low personal accomplishment scores. As noted in our discussion, other studies indicate that poor perception of personal accomplishment by individual precedes signs of emotional exhaustion. Thus, by the time emotional exhaustion is identified, burnout might be already underway. The respondents in our study had very low scores of perception of their personal accomplishment (<=33).

In response to you comment, we revised our results and discussion to make sure our results and their interpretation are not overstated or overreaching. Please see Results and Discussion. 

15. The list of stressors with ‘relatively higher scores’ includes nearly all of them (10 of 13), and 8/13 score 3 on the 5 point scale. As scores are rounded to whole numbers this suggests most were at the scale midpoint, not ‘high’ scores.

AU: Thank you for the comment. We replaced the phrase “relatively higher scores” with “moderate”. Please see Abstract and Results (qualitative data collection on perception of level of stress).

16. Overall, the data do not give strong evidence of problems throughout the cohort. More interestingly perhaps, variation in the data suggests that some individuals may be experiencing difficulties, so targeted solutions matched to individual needs may be required.

AU: As indicated in our introduction, house officers are veterinarians who are potentially under stress during training and our aim was to investigate if their mental wellness is optimal given that veterinary students and other practicing veterinarians have been reported to have elevated psychological stress compared the general population. You are correct, our data does not give strong evidence that that our cohort had higher levels of psychological stress compared to other practicing veterinarians (which was surprising but desirable). We indicated in our discussion proposed ways to offer resources, in this case a faculty member to resolve personal/service concerns. As indicated, one of the major drawback is that our study is preliminary study and has limited external validity, but it might serve as a basis for our/other house officers programs. 

17. Conclusions

18. The paper focuses on problems and argues for interventions but the evidence is thin. There is little said about the null findings for several variables or that, where others exceeded cut-offs, it was never by much. There are considerable problems – logical and statistical – with saying that a score of 49.1 is fine but 50.1 is problematic, for instance.

AU: Thank you for that comment and your point is well taken. Please see our comment as to why we had focused on “problems” in response to comment #16.

Other research recommends focusing on the aspects that improve wellness. In response to your comment, we indicated in the discussion about considering studies that focus on positive aspects, such as measure of happiness. Please see Discussion and relevant references.

19. As the PHQ is described, it’s not possible to be ‘not at all’ depressed; at best people have to be ‘mildly’ depressed. Depression is a serious diagnosable mental health condition, so telling participants that they are ‘mildly depressed’ based on these scores could cause problems in itself.

AU: You point is well taken. As indicated in the materials and methods, the PHQ-9 is not a screening tool for depression, but it is validated and used to monitor severity if depression or response to treatment and thus the results in our study should be interpreted as such. The description and use of PHQ-9 were made explicit in our first submission.

20. Claims that participants ‘consistently experienced high levels of burnout’ do not match the data, from which burnout does not appear to be extreme or widespread.

AU: Please see our response for item #14 for the response on why we considered our cohort to have high levels of burnout (high emotional exhaustion scores and low scores of personal accomplishment scores). We hope this clarifies the reviewer’s concerns regarding our conclusions on burnout.

21. The authors suggest that ‘burnout was associated with stressors such as…’ but no tests of association (e.g. correlation) are presented.

AU: Thank you for the comment. We replaced the word “correlation” with the phrase “might be related to..”

22. Conclusions about possible intervening variables - coping and resilience, and wider implications e.g. transfer to working environments, are not part of the study.

AU: We deleted the information regarding wider implications, coping, and resilience. Thank you.

23. Although there is little evidence of problems, the wider literature is used to argue that there are in fact problems which need to be solved. Scale ranges suggest that some participants may be experiencing problems but this does not show which interventions are required or by whom. Targeted approaches focused on needs would seem preferable. The suggested solutions are not based on the study findings.

AU: Please see our responses to your comments in #16 and #18 above.

24. Clarity

25. It’s well written and readable, except that the use of acronyms in the results section means constantly referring back to what each one means. There’s also a constant need to refer back to information on score thresholds to work out what mean scores indicate.

AU: We responded to the scoring of the content analysis and changed the phrase “relatively higher scores” with “moderate”. Please see responses to comment #15 above.

We revised the manuscript and minimized acronyms to reduce distraction to the reader. 

26. Summary

27. Overall, then, this is a small-scale study in one training institution which does not allow conclusions to be drawn about a profession, or even veterinary training, as a whole. The findings are over-stated and focus too much on the negative, and the suggested interventions are unrelated to the study findings. While professional well-being in veterinarians is interesting and important, the conclusions should be carefully aligned with the findings.

AU: Thank you for the comments. Below is the summarized response for the above comment:

1. We revised the manuscript to ensure our conclusions are not overreaching or overstated.

2. As pointed out in the introduction, this is a preliminary study and might serve as a basis for future studies.

3. We also acknowledged why our study focused on the psychological stress based on previous studies in veterinary students and practicing veterinarians. We also acknowledged that further studies should also focus on aspects of resident training that improve mental health and wellness.

4. We also included results of multivariate analysis as you recommended and discussed the results. 

Reviewer #3: Comments for the Authors:

Thank you for the opportunity to review your manuscript. Wellbeing of veterinarians is an important scientific topic and deserves to be investigated.

After reading your manuscript one is confused what the purpose was of carrying out the study in two consecutive school-years? What made you think that veterinarians’ wellbeing may be changed in one year? What happened in that period that might approve this study design? Another question is how many people from the 2017 and 2018 samples are actually the same persons? We don’t know anything about that after reading your description of the results? Further, as the samples from 2017 and 2018 are not dependent nor independent but mixed, there is no statistical test that might compare the respondents wellbeing in 2017 and 2018. In conclusion, you have chosen a wrong method to fulfill the purpose of the study. Therefore, unfortunately I have to recommend the editor to reject your paper.

AU: Thank you for taking time to review our manuscript. Please see our responses to your comments:

1. We included the reason for including 2 cohorts from 2017 and 2018. We included the cohorts to increase our sample size for the cross-sectional study. We indicated this in the Materials and Methods. 

2. We included the description of the House Officers Program, including the length of the residency programs and fellowships/internships (Please see the Introduction). Whereas the fellows/interns respondents were different in 2017 and 2018 because they are 1 year-programs, some of residents (unless they were in their final year of the residency program) potentially responded to the surveys in both 2017 and 2018. We did not track the residents who might have responded to the survey in 2017 and 2018 to maintain anonymity of the respondents. Anonymity was important because the responses are considered personal medical information. 

3. We also included multivariate analysis determining variables associated with the survey scores as suggested and discussed the results.

---

## [Decision Letter · Decision Letter 1]

19 Apr 2021

PONE-D-20-17459R1

A preliminary study on assessment of wellbeing among veterinary medical house officers

PLOS ONE

Dear Dr. Chigerwe,

Thank you for submitting your manuscript to PLOS ONE. After careful consideration, we feel that it has merit but does not fully meet PLOS ONE’s publication criteria as it currently stands. Therefore, we invite you to submit a revised version of the manuscript that addresses the points raised during the review process.

We look forward to receiving your revised manuscript.

Kind regards,

Jenny Wilkinson, PhD

Academic Editor

PLOS ONE

Additional Editor Comments (if provided):

Thank you for your revisions and responses to reviewers; one of the original reviewers have [provide some further comments. I invite you to consider these comments and provide responses, and if necessary, revisions to the manuscript. These comments particular focus on clarity of your work and on ensuring that findings are not overstated.

Reviewers' comments:

Reviewer's Responses to Questions

**Comments to the Author**

1. If the authors have adequately addressed your comments raised in a previous round of review and you feel that this manuscript is now acceptable for publication, you may indicate that here to bypass the “Comments to the Author” section, enter your conflict of interest statement in the “Confidential to Editor” section, and submit your "Accept" recommendation.

Reviewer #1: All comments have been addressed

Reviewer #2: (No Response)

Reviewer #3: All comments have been addressed

2. Is the manuscript technically sound, and do the data support the conclusions?

Reviewer #1: Yes

Reviewer #2: No

Reviewer #3: Yes

3. Has the statistical analysis been performed appropriately and rigorously? 

Reviewer #1: Yes

Reviewer #2: No

Reviewer #3: Yes

4. Have the authors made all data underlying the findings in their manuscript fully available?

Reviewer #1: Yes

Reviewer #2: No

Reviewer #3: Yes

5. Is the manuscript presented in an intelligible fashion and written in standard English?

Reviewer #1: Yes

Reviewer #2: Yes

Reviewer #3: Yes

6. Review Comments to the Author

Reviewer #1: Thank you for addressing the comments. I think the manuscript is in good shape to be recommended for publication.

Reviewer #2: 1. The focus is on veterinary house officers and the potential stressors they face, which is interesting. It would be helpful to contextualise it with information on other health professionals, but this information doesn't appear until the Discussion. It would be better in the introduction to set the scene for the study. There is good information on the training programme and support available. It would be interesting to know the extent to which house officers use the supports.

2. Some clarification is needed in the method section.

3. Internal consistencies are needed for each scale/subscale based on the data for this study – put this in the Method not the Results. Where reliabilities for the entire measure and for subscales are given, clarify which was used in the analysis and why. For all scales, state how many items were in each subscale, what the scale anchors were, and whether scores were calculated as the sum or mean of items.

4. Expand the data analysis section. Which variables met parametric assumptions, which didn’t? Which analysis were carried out using which methods? In some cases medians are given, suggesting non-parametric data, but MANOVA was used. Consistency and clarity in the analysis are required.

5. The MBI section states that it ‘gives a context for why burnout could have possibly occurred’ but this is not part of the study – the MBI does not measure the causes of burnout e.g. work demands.

6. For burnout, avoid calling all subscales ‘burnout’. For instance, p.6: “The emotional exhaustion subscale assesses … indicating low, average and high levels of burnout, respectively” should read “… indicating low, average and high levels of emotional exhaustion, respectively.” Same for depersonalisation and personal accomplishment. This also applies to the Results section e.g. footnote to Table 1.

7. As noted in my earlier review I am concerned that at best a person must be ‘mildly depressed’ on the PHQ-9. How were ethical requirements for avoiding harm managed when giving participants feedback?

8. For the SF-8, how many items were there for the physical and mental scales? What were the scale reliabilities and ranges? How were scores ‘standardised’?

9. The rating scales for stressors are quantitative not qualitative as numerical scores are given. The free text answers are, of course, qualitative, but not analysed here.

10. Page 3 states that there were 48 training positions in each year. Two years of data were collected, and p. 9 states that there were 175 potential participants in one year and 119 in the next year. This does not fit the description of the programme.

11. The demographic information should be summarised in a Table.

12. Comparing the two years was not the focus of the study. It would be simpler to state, near the start of the results, that there were no meaningful differences between the two cohorts and to carry out all analyses on the combined datasets.

13. Where medians and ranges are given, include the IQR. Alternatively, use mean and SD if suitable.

14. The findings and Discussion over-state the level of distress identified.

15. Anxiety: the medians show ‘mild’ anxiety (p. 6).

16. Burnout. While emotional exhaustion was high, depersonalisation was average, and feelings of personal accomplishment were on the boundary between low and average. This suggests fatigue/weariness rather than problematic levels of the much more severe syndrome of burnout.

17. PHQ-9 ‘mild’ depression - as mentioned above it’s not possible to be better than ‘mildly depressed’. This scale is, by its nature (clinical assessment) designed to show distress. Median scores were around 7 showing, at worst, ’moderate’ levels of depression.

18. No weight can be placed on the gender comparisons as there were so few males; married participants were also under-represented.

19. The analysis is unclear. What is meant by ‘when gender [or marital status] was considered as the independent variable, … scores were positively correlated with… ‘? This suggests regression with demographics as control variables (not independent variables) but then the reference to correlation doesn’t fit, and the data analysis section mentions MANOVA – again, not correlation.

20. The paragraph on p. 12 about the QoL measure is nearly incompressible. As this scale was analysed in terms of two scales – physical and mental – that should be the level of analysis rather than the ‘dimensions’ which are the separate items.

21. While some scores on the QoL measure were ‘below 50’ they were not far below. A rigid cut-off of 50 provides no information on the variance or standard errors associated with this cut-off. The data show, instead, that the participant group is close to ‘normal’, as would be expected of a hard-working and busy professional group.

22. The stress data show that participants endorsed most items, which is to be expected as the items were chosen because they are known sources of stress to working professionals. There’s nothing to say these were problematic stressors for participants.

23. The discussion over-states the levels of distress in this group. Participants appear, by and large, to be doing rather well, although busy and probably at times feeling overwhelmed – that would be normal in a demanding professional programme. Some individuals may be experiencing genuine distress, and may need help and support, but as regards the group, or house officers in general, the data don’t show that they’re overly distressed or unwell. In addition, the study can’t say what’s causing any fatigue/emotional exhaustion – its’ a cross-sectional study and there’s no demonstrated link between the stressor items and the mental health hones. Nothing can’t show causation, the information on work demands is unlinked to the wellbeing data, and some demographic subgroups are too small. The study does not tell us what helps and what doesn’t hep this sample of house officers to manage their wellbeing. Nothing can be said about training as a solution, or that ‘burnout is inevitable’ (p. 18). Much more caution is needed in interpreting the findings.

Reviewer #3: Thank you again for the opportunity to review the article. You have corrected everything that was required.

7. PLOS authors have the option to publish the peer review history of their article (what does this mean?). If published, this will include your full peer review and any attached files.

Reviewer #1: No

Reviewer #2: No

Reviewer #3: No

---

## [Author Response · Author response to Decision Letter 1]

25 May 2021

Dr. Wilkinson

Academic Editor

Thank you for reviewing our manuscript. Below are our responses to the reviewers’ comments.

Academic Editor Comments

Thank you for your revisions and responses to reviewers; one of the original reviewers have provide some further comments. I invite you to consider these comments and provide responses, and if necessary, revisions to the manuscript. These comments particular focus on clarity of your work and on ensuring that findings are not overstated.

AU: We addressed the reviewer’s concerns regarding clarity and conclusions from our study.

Reviewer #1: Thank you for addressing the comments. I think the manuscript is in good shape to be recommended for publication.

AU: Thank you.

Reviewer #2: 1. The focus is on veterinary house officers and the potential stressors they face, which is interesting. It would be helpful to contextualise it with information on other health professionals, but this information doesn't appear until the Discussion. It would be better in the introduction to set the scene for the study. There is good information on the training programme and support available. It would be interesting to know the extent to which house officers use the supports.

AU: Thank you for your thorough review of our manuscript. Below are our responses to your comments and suggestions. To make it easier for you to identify specific responses to your comments, we highlighted the changes in yellow in the body of the manuscript.

We focused on veterinary medicine in the introduction because several studies are available for practicing veterinarians and there are also perceived differences in risk factors for mental health issues among various health professionals (for instance, veterinarians versus physicians).

2. Some clarification is needed in the method section.

3. Internal consistencies are needed for each scale/subscale based on the data for this study – put this in the Method not the Results. Where reliabilities for the entire measure and for subscales are given, clarify which was used in the analysis and why. For all scales, state how many items were in each subscale, what the scale anchors were, and whether scores were calculated as the sum or mean of items.

AU: We clarified your comments and suggestions as follows:

a. The internal consistency Cronbach alphas we provided in the materials and methods for each instrument (Maslach, GAD-7, PHQ-9 and SF-8) were the original alphas for validation of the instruments (as you requested in the first review). Whereas, the Cronbach’s alphas for the scales/subscales indicated in the results are Cronbach’s alphas from our study. We think it is desirable to indicate the Cronbach’s alphas from our study in the results section so that the reader can relate these to the original alphas reported when the instruments were developed (materials and methods). We had provided these values, but in response to your comment, we clarified the source of the Cronbach’s alphas in the materials and methods (original) and results (from our study).

b. We indicated whether scale or subscales Cronbach’s alphas were used in our analysis, and reasons for the choice, namely:

Maslach Burnout Inventory (MBI)– we used the 3 subscales emotional exhaustion, depersonalization, and personal accomplishment. The subscales better describe the context of why burnout might have occurred.

Generalized Anxiety Disorders (GAD-7) and Patient Health Questionnaire (PHQ-9): The instruments have no subscales therefore whole scale alphas were used.

Quality of Life: Subscales mental component summary (MCS) and physical mental summary (PCS) are more relevant than the whole scale because they might guide specific interventions 

to improve the mental or physical component of quality of life.

Please see “data analysis” section for the changes.

c. We had indicated the number of items in each instrument in our previous version of the manuscript. Please see materials and methods.

d. We had indicated the scale anchors for GAD-7 in our previous manuscript version. We included the anchors for scores 1 (several days) and 2 (more than half the days) for PHQ-9. The anchors for MBI and SF-8 (quality of life) depends on the item, thus, it will be easier for the reader to look at the instrument for specific anchors. As indicated in our previous version of the manuscript, we cannot provide the specific items for the MBI and SF-8 because they are copyrighted. Please see materials and methods for the changes.

e. We clarified how the scores were calculated for each instrument. Please note for the MBI we only indicated the number of items used to obtain the scores for the subscales. As indicated in our previous version of the manuscript, a software standardizes the responses against the referent United States population for the SF-8. Please see materials and methods for the changes.

4. Expand the data analysis section. Which variables met parametric assumptions, which didn’t? Which analysis were carried out using which methods? In some cases medians are given, suggesting non-parametric data, but MANOVA was used. Consistency and clarity in the analysis are required.

AU: All variables with mean reported followed a normal distribution whereas those with medians reported were not normally distributed. 

You are correct that a MANOVA (assumes normal distribution) was used and yet some of the data points for some variables were not normally distributed. Because there isn’t a straightforward way in the software we used to run a non-parametric MANOVA, we did the following during the initial data analysis to make sure that running a MANOVA was appropriate:

a. We checked for assumptions for using MANOVA to be met by checking whether the residuals were normally distributed and homogeneity of variances was present. MANOVA is fairly robust for underlying distributions and is not sensitive to violations of multivariate normality when the data does not contain outliers.

b. To ensure the above in (a), we also ran a MANOVA on ranks and re-analyzed the data and arrived at the same conclusions.

We included the clarification summary of relevant information in the data analysis section.

5. The MBI section states that it ‘gives a context for why burnout could have possibly occurred’ but this is not part of the study – the MBI does not measure the causes of burnout e.g. work demands.

AU: The “context on why burnout could have possibly occurred” refers to the subscales (emotional exhaustion, depersonalization, personal accomplishment) and not the causes of burnout. Each of the subscales will have different specific reasons for the score magnitude and these have to be investigated. Our statement did not intent to refer to the causes of burnout. We hope this clarifies the reviewer’s comment.

6. For burnout, avoid calling all subscales ‘burnout’. For instance, p.6: “The emotional exhaustion subscale assesses … indicating low, average and high levels of burnout, respectively” should read “… indicating low, average and high levels of emotional exhaustion, respectively.” Same for depersonalisation and personal accomplishment. This also applies to the Results section e.g. footnote to Table 1.

AU: We respectfully disagree with the reviewer regarding this comment and interpretation. High emotional exhaustion scores alone, or high levels of depersonalization alone or low scores of personal accomplishment alone are interpreted as indicating burnout. Additionally, a combination of scores for the subscales outside the ranges is also interpreted as indicating burnout. Thus for instance, one cannot conclude a group of individuals who have high emotional exhaustion scores, but within reference depersonalization and personal accomplishment scores are not experiencing burnout. Our argument is based on how the scores are interpreted in the Maslach Burnout Inventory Manual. We did not change the interpretation in the manuscript. We hope the reviewer is comfortable with this comment.

7. As noted in my earlier review I am concerned that at best a person must be ‘mildly depressed’ on the PHQ-9. How were ethical requirements for avoiding harm managed when giving participants feedback?

AU: Thank you for the important comment. All identifiers for the respondents were removed and anonymity of respondents was protected before analysis and access by the investigators. The data was reported as median or means for the whole group of house officers. In the event of a house officer requesting individual scores, there were plans to recommend them to discuss the results with Academic and Staff Assistance Program or personal therapist. In our study, none of the house officers requested individual results. It is also important to note that the PHQ-9 is not a screening tool for depression but it is used to monitor severity of depression and response to treatment as noted in the Materials and Methods. Additionally, during administration of instruments, the purpose of each instrument was described. All questions were not delivered in a forced format, and respondents could skip questions.

8. For the SF-8, how many items were there for the physical and mental scales? What were the scale reliabilities and ranges? How were scores ‘standardised’?

AU: a. Some items assess both physical and mental components whereas some assess mental component only and others assess the physical component only. 

For instance the item: 

a. “Overall, how would you rate your health during the past 4 weeks?” assesses both mental and physical components.

Compared to the following items:

a. During the past 4 weeks, how much did physical health problems limit usual physical activities (such as walking or climbing stairs)?” assess the physical component.

b. During the past 4 weeks, how much have you been bothered by emotional problems (such as feeling anxious, depressed or irritable)?” assesses the mental component summary.

We had indicated the scale reliabilities for the mental and physical component summaries. Please also see our response above to why we chose to report the reliability for the mental and physical component summaries. Please see materials and methods under data analysis. 

As indicated in the materials and methods, responses for each item for each individual respondent are standardized using a software (Optum Pro CoRE). Standardizing refers to converting the responses to scores for the eight dimensions and comparing them to a referent United States population. Please see materials and methods under ‘Quality of life” section.

9. The rating scales for stressors are quantitative not qualitative as numerical scores are given. The free text answers are, of course, qualitative, but not analyzed here.

AU: Content analysis (free text) were analyzed as stated in the previous version of the manuscript. Please see under “data analysis” and results section under “quantitative and qualitative data collection on perception of level of stress”

We added the word “qualitative” on the subheading “quantitative and qualitative data collection on perception of level of stress” to clarify your comment.

10. Page 3 states that there were 48 training positions in each year. Two years of data were collected, and p. 9 states that there were 175 potential participants in one year and 119 in the next year. This does not fit the description of the programme.

AU: The 48 refers to the maximum positions available each year. To clarify, the number of positions available differs each year due the following reasons:

a. A program may hire more than one house officer per year whereas some programs may hire a house officer every other year or hire different number each year (for instance 2 house officers in one year and 1 house officer the following year). Thus in a year where all programs are hiring their maximum number of allocated positions approximately 175 participants can be reached; mean of 43 (175/4) house officers per year -1st, 2nd, 3rd and 4th years in 2017. As indicated the in 2018 only 119 participants were available indicating that some of the house officers graduated and other programs did not hire any house officers or hired less house officers.

b. Fellowships and internships are 1 yearlong whereas residency programs are 3-4 years.

To clarify we added the word “maximum” before “48” and also added that the number of house officers hired varies depending on the specialty. Please see introduction under information on the residency program.

11. The demographic information should be summarised in a Table.

AU: Thank you. We summarized the demographic information in a table (Table 1). We changed the identification of the other tables accordingly. We deleted the sentences with the information on demographic information in the body of the manuscript. 

12. Comparing the two years was not the focus of the study. It would be simpler to state, near the start of the results, that there were no meaningful differences between the two cohorts and to carry out all analyses on the combined datasets.

AU: While we agree with your comment, we wanted also to demonstrate that the despite the cohorts being dissimilar, the scores were not significantly different. One of the reviewers had also indicated to leave the data as indicated for 2017 and 2018. 

In response to your comment, we included calculation of combined scores for GAD-7 and PHQ-9, and a column for the combined years form MBI (Table 2) and SF-8 (Table 3). We hope this is acceptable to the reviewer.

13. Where medians and ranges are given, include the IQR. Alternatively, use mean and SD if suitable.

AU: Strictly speaking the range is the difference between the minimum and maximum data points and we had provided both the minimum and the maximum data points in the previous version of the manuscript. We do not think it is necessary to provide both the range and IQR as measures of dispersion for the median. In response to your comment, we included IQR as the measure of dispersion for the median because the range is biased and prone to extremely small and large measurements. Please see results.

We had provided mean and standard deviation for the measurements that were normally distributed in our previous manuscript.

14. The findings and Discussion over-state the level of distress identified.

AU: As we indicated in our previous version, we emphasized the limitations of our study in our discussion mainly focusing on the limited external validity, and overrepresentation of various demographics (biased towards unmarried and or female respondents). Please see also our respondents to your comments regarding interpretation of the survey instruments. We re-read our discussion to make sure our conclusions do not overreach beyond our study results and interpretations.

15. Anxiety: the medians show ‘mild’ anxiety (p. 6).

AU: That is correct that our results interpretation of scores of 5-9 are indicative of mild anxiety (Page 6). Median score (for both years) from our study was 5, and that is how we interpreted the results. 

16. Burnout. While emotional exhaustion was high, depersonalisation was average, and feelings of personal accomplishment were on the boundary between low and average. This suggests fatigue/weariness rather than problematic levels of the much more severe syndrome of burnout.

AU: The scores for personal accomplishment (PA) were 32 (2017), 33 (2018) and 32 (combined). The interpretation for PA scores as indicated in Table 2 are: ≥ 40, 39–34 and ≤ 33 indicate low, average, and high levels of burnout, respectively. The scores are ≤ 33 indicate high levels of burnout. Please see also our responses to #6 regarding interpretation of the Maslach Burnout Inventory. We hope this clarifies the reviewer’s comments. 

17. PHQ-9 ‘mild’ depression - as mentioned above it’s not possible to be better than ‘mildly depressed’. This scale is, by its nature (clinical assessment) designed to show distress. Median scores were around 7 showing, at worst, ’moderate’ levels of depression.

AU: As mentioned in the materials and methods, the PHQ-9 is not used for screening for depression but it is used to monitor depression or response to treatment. Therefore using the adverb “mildly” implies diagnosis of depression from a screening test/clinical examination. Although this may sound technical, it’s an important differentiation and hence we maintained using the adjective “mild” as recommended by the survey instrument.

18. No weight can be placed on the gender comparisons as there were so few males; married participants were also under-represented.

AU: We agree with the reviewer and we had indicated that the results need to be interpreted with caution in the previous version of the manuscript. We retained the comments regarding interpretation of the results regarding gender.

19. The analysis is unclear. What is meant by ‘when gender [or marital status] was considered as the independent variable, … scores were positively correlated with… ‘? This suggests regression with demographics as control variables (not independent variables) but then the reference to correlation doesn’t fit, and the data analysis section mentions MANOVA – again, not correlation.

AU: 

a. To clarify we used the term “explanatory” variable rather than “independent” variable based on your comment.

b. “When gender (or marital status) was considered as the independent variables” refers to when either is considered the explanatory variable for the scores. Because of the over-representation of females and unmarried respondents we chose to assess each significant explanatory variable (gender or marital status) one at a time and evaluate its effect on the scores of the 4 instruments. In response to your comment, we simplified, the analysis and re-ran the MANOVA with both gender and marital status in the model. The correlation coefficients were similar and the conclusions remained the same. Please see results.

c. It is important to note that each respondent has scores on 4 instruments (GAD-7, PHQ-9, MBI, and SF-8). However, those scores from each respondent on each instrument are not independent, i.e they are correlated. In addition to analyzing variance (similar to ANOVA) between the scores as a function of the explanatory variables (gender and marital status and age in our case) MANOVA also allows investigation of the correlations (strength and direction of the correlation) among the scores from the various instruments while maintaining a single type 1 error (no adjustment for multiple comparisons). The correlations are important information in that they give an idea of the relationship between the scores from the different instruments. For instance, in our study, emotional exhaustion scores were positively correlated with PHQ-9 scores (r=0.59). We hope this clarifies to the reviewer the reason for inclusion of correlations. 

20. The paragraph on p. 12 about the QoL measure is nearly incompressible. As this scale was analysed in terms of two scales – physical and mental – that should be the level of analysis rather than the ‘dimensions’ which are the separate items.

AU: We understand the reviewer’s comment especially given that we are not able to publish the items, which would make it clear. While the physical and mental component summary are what we used to make interpretations, the scores are derived from the 8 dimensions. We think it is important to include the scores for each dimension as they might help the approaches to interventions when necessary i.e does the intervention need to focus specifically on certain dimensions to improve mental or physical health? For instance, the following item assess the “Social Functioning” dimension: “During the past 4 weeks, how much did your physical health or emotional problems limit your usual social activities with family or friends?” Social functioning is dimension that is affected by either physical, mental or both mental and physical health. Thus, if the score for this dimension is poor, it might be recommended to improve both mental and physical components of quality of life. In contrast the following item assesses physical functioning dimension: “During the past 4 weeks, how much did physical health problems limit your usual physical activities (walking, climbing stairs)?” Intervention for this dimension might be focused on physical component of quality of life.

We hope this context is acceptable to the reviewer.

21. While some scores on the QoL measure were ‘below 50’ they were not far below. A rigid cut-off of 50 provides no information on the variance or standard errors associated with this cut-off. The data show, instead, that the participant group is close to ‘normal’, as would be expected of a hard-working and busy professional group.

AU: Thank you for the comment. As indicated above the scores from the responses are converted to scores that are compared to a referent American population. In the original validation of the quality of life instrument, the reported standard deviation was 10.

22. The stress data show that participants endorsed most items, which is to be expected as the items were chosen because they are known sources of stress to working professionals. There’s nothing to say these were problematic stressors for participants.

AU: We agree with the reviewer’s comments. We only made comments on those items that might be associated with burnout. Furthermore, these items might be only be relevant to our study (for instance physical environment of the teaching hospital) or specific specialty (for instance patient management is irrelevant for an anatomic pathology residency program.

23. The discussion over-states the levels of distress in this group. Participants appear, by and large, to be doing rather well, although busy and probably at times feeling overwhelmed – that would be normal in a demanding professional programme. Some individuals may be experiencing genuine distress, and may need help and support, but as regards the group, or house officers in general, the data don’t show that they’re overly distressed or unwell. In addition, the study can’t say what’s causing any fatigue/emotional exhaustion – its’ a cross-sectional study and there’s no demonstrated link between the stressor items and the mental health hones. Nothing can’t show causation, the information on work demands is unlinked to the wellbeing data, and some demographic subgroups are too small. The study does not tell us what helps and what doesn’t hep this sample of house officers to manage their wellbeing. Nothing can be said about training as a solution, or that ‘burnout is inevitable’ (p. 18). Much more caution is needed in interpreting the findings.

AU: Our results focused on burnout for which have evidence of its presence (high emotional exhaustion and low personal accomplish scores). We made a comment that the scores on anxiety and depression were low and we are in fact happy about it (and we gave reasons why are results are different from other practicing veterinarians). Burnout results from an individual being overstretched in their work place and if the source of burnout is removed or reduced, the stress experienced by an individual is reduced whereas depression affects one’s holistic life (work and outside work). Burnout can predispose to depression and in our study, it is a concern that need to be addressed.

We did not investigate causality but rather associations between explanatory variables and the scores and we were cautious with our interpretations as indicated. The recommendations we indicate in our study are based on other studies in veterinary/medical fields and further studies are required to make such comments. As you point out this is a cross-sectional study, therefore only associations are appropriate.

We re-read the discussion to make sure our conclusions are not overreaching.

Reviewer #3: Thank you again for the opportunity to review the article. You have corrected everything that was required.

AU: Thank you.

---

## [Editor Report · Decision Letter 2]

31 May 2021

A preliminary study on assessment of wellbeing among veterinary medical house officers

PONE-D-20-17459R2

Dear Dr. Chigerwe,

We’re pleased to inform you that your manuscript has been judged scientifically suitable for publication and will be formally accepted for publication once it meets all outstanding technical requirements.

Kind regards,

Jenny Wilkinson, PhD

Academic Editor

PLOS ONE

Additional Editor Comments (optional):

Thank you for your detailed responses and manuscript revisions; these have satisfactorily addressed reviewer comments.
---

## [Editor Report · Acceptance letter]

16 Jun 2021

PONE-D-20-17459R2 

A preliminary study on assessment of wellbeing among veterinary medical house officers 

Dear Dr. Chigerwe:

I'm pleased to inform you that your manuscript has been deemed suitable for publication in PLOS ONE. Congratulations! Your manuscript is now with our production department. 

Kind regards, 

on behalf of

Dr Jenny Wilkinson 

Academic Editor

PLOS ONE